# Sex Differences in Repolarization Markers: Telemonitoring for Chronic Heart Failure Patients

**DOI:** 10.3390/jcm12144714

**Published:** 2023-07-16

**Authors:** Federica Moscucci, Susanna Sciomer, Silvia Maffei, Antonella Meloni, Ilaria Lospinuso, Myriam Carnovale, Andrea Corrao, Ilaria Di Diego, Cristina Caltabiano, Martina Mezzadri, Anna Vittoria Mattioli, Sabina Gallina, Pietro Rossi, Damiano Magrì, Gianfranco Piccirillo

**Affiliations:** 1Department of Internal Medicine and Medical Specialties, Policlinico Umberto I, Viale del Policlinico n. 155, 00161 Rome, Italy; 2Dipartimento di Scienze Cliniche, Internistiche, Anestesiologiche, Cardiovascolari, “Sapienza” University of Rome, 00185 Rome, Italy; susanna.sciomer@uniroma1.it (S.S.); lospinuso.i@gmail.com (I.L.); myriam.carnovale@uniroma1.it (M.C.); andrea.corrao@uniroma1.it (A.C.); ilaria.didiego@uniroma1.it (I.D.D.); cristina.caltabiano@uniroma1.it (C.C.); martina.mezzadri@uniroma1.it (M.M.); gianfranco.piccirillo@uniroma1.it (G.P.); 3Endocrinologia Cardiovascolare Ginecologica ed Osteoporosi, Fondazione G. Monasterio CNR-Regione Toscana, 56124 Pisa, Italy; silvia.maffei@ftgm.it; 4Department of Radiology, Fondazione G. Monasterio CNR-Regione Toscana, 56124 Pisa, Italy; antonella.meloni@ftgm.it; 5Surgical, Medical and Dental Department of Morphological Sciences Related to Transplant, Oncology and Regenerative Medicine, University of Modena and Reggio Emilia, 42121 Modena, Italy; annavittoria.mattioli@unimore.it; 6Department of Neuroscience, Imaging and Clinical Sciences, Institute of Advanced Biomedical Technologies, “G. D’Annunzio” University, 66100 Chieti, Italy; s.gallina@unich.it; 7Arrhythmology Unit, Fatebenefratelli Hospital Isola Tiberina—Gemelli Isola, 00186 Rome, Italy; rossi.ptr@gmail.com; 8Dipartimento di Medicina Clinica e Molecolare, S. Andrea Hospital, “Sapienza” University of Rome, 00185 Rome, Italy; damiano.magri@uniroma1.it

**Keywords:** myocardial repolarization, Tend, sex differences, chronic heart failure telemonitoring

## Abstract

Aging and chronic heart failure (CHF) are responsible for the temporal inhomogeneity of the electrocardiogram (ECG) repolarization phase. Recently, some short period repolarization–dispersion parameters have been proposed as markers of acute decompensation and of mortality risk in CHF patients. Some important differences in repolarization between sexes are known, but their impact on ECG markers remains unstudied. The aim of this study was to evaluate possible differences between men and women in ECG repolarization markers for the telemonitoring of CHF patients. Method: 5 min ECG recordings were collected to assess the mean and standard deviation (SD) of the following variables: QT end (QTe), QT peak (QTp), and T peak to T end (Te) in 215 decompensated CHF (age range: from 49 to 103 years). Thirty-day mortality and high levels of NT-pro BNP (<75 percentile) were considered markers of decompensated CHF. Results: A total of 34 patients (16%) died during the 30-day follow-up, without differences between sexes. Women showed a more preserved ejection fraction and higher LDL and total cholesterol levels. Among female patients, implantable cardioverter devices, statins, and antiplatelet agents were less used. Data for Te mean showed increased values among deceased men and women compared to survival, but Te_SD_ was shown to be the most reliable marker for CHF reacutization in both sexes. Conclusion: Te_SD_ could be considered a risk factor for CHF worsening and complications for female and male patients, but different cut offs should be taken into account. (ClinicalTrials.gov number, NCT04127162.)

## 1. Introduction

Acutely decompensated chronic heart failure (CHF) is among the most frequent causes of hospitalization in Western countries [1], presenting a high mortality rate and a serious burden for patients’ quality of life and for cost charges in the health systems. Mortality for CHF depends on sex, age, reacutization frequency, comorbidities, and severity of the CHF itself. According to recent evidence, proper early risk stratification and management could help reduce the burden of this chronic disorder [2]; thus, telemonitoring is promising and undelayable, using accurate prognostic markers for optimal risk stratification and early intervention for high-risk patients.

Aside from blood tests and echocardiography, electrocardiogram (ECG) is one of the essential tools in the investigation, management, and follow-up of heart failure among both in-patients and out-patients [3,4,5,6]. In particular, ECG is readily available, inexpensive, and could be a useful indicator of re-exacerbation in CHF patients [7]. However, gender-specific data of the prognostic value of ECG measurements are lacking.

As evidenced by many recent studies [8,9,10,11,12,13,14], heart failure in women shows extremely peculiar characteristics. In fact, a later age at the diagnosis [8] affects more women than men; less women receive implantable devices and resynchronization therapy [8,9]. These characteristics are strictly related to the protective role of estrogens during the fertility age; these hormones have been demonstrated to enhance the endothelial nitric oxide synthase (eNOS) role in the endothelium layer, ameliorating the flux-mediated vasodilatation, and preventing atherosclerotic and ischemic damage [10,11,12]. The physiopathology of the ischemic damage is strictly related to the endothelial dysfunction and micro-vessel damage [15,16,17], which, in women, lead to a CHF with a more preserved ejection fraction than in men [8]. For this reason, symptoms, clinical course, therapeutic response, and prognosis [13,18,19,20] (Table 1) are peculiar, and the repolarization ECG patterns [21,22] and specific markers of clinical curse and prognosis should also be evaluated [13,14,16,17].

Specific diagnostic and therapeutic interventions, together with an education program to enhance clinicians’ and women’s awareness, are desirable [23].

All in all, these patients, especially the oldest ones, frequently show an increased incidence of arrhythmias and electrical conduction disturbances (atrial fibrillation, premature beats, bundle branch block, etc.), so they are often excluded from trials [8], despite being the group with the more impaired quality of life and more severe lack of independence in everyday life activities.

Therefore, we studied the repolarization variables (thus, sinus rhythm was not necessary) in order to better understand sex differences in heart failure repolarization markers and to better estimate arrhythmic risk, and those variables were specifically customed for telemonitoring devices.

## 2. Methods and Materials

### 2.1. Participants and Protocol

A total of 215 consecutive symptomatic outpatients with acutely decompensated chronic heart failure (adCHF) were enrolled at admission to the Geriatric, Internal Medicine and Cardiovascular divisions of Policlinico Umberto I in Rome from January 2019 to October 2022, with an enrollment interruption of 14 months from March 2020 due to the SARS-CoV2 pandemic. Decompensated CHF were defined as patients with at least one symptom or sign compatible with a reacutization and a previous documented history of CHF, following European Society of Cardiology guidelines (2016 and 2021) [24,25]. At the time of hospitalization, all patients underwent full clinical history, physical examination, standard electrocardiogram (ECG) evaluation and transthoracic echocardiography, 5 min of II lead ECG (Miocardio EventTM, Rome, Italy) recording, and a blood sample for routine plasma tests (serum electrolytes, creatinine, urea, ultra-sensible troponin T, C-reaction protein -CRP-, and NT-pro Brain Natriuretic Peptide -NT-pro-BNP, etc.). Among the twenty-four hours before the planned hospital discharge, the patients repeated the 5 min ECG recording and the NT-pro BNP plasma level dosage. To assess the creatinine clearance, the Cockcroft–Gault formula was used.

### 2.2. Off-Line Data Analysis

A custom-designed card (National Instruments USB-6008; National Instruments, Austin, TX) was used to acquire and digitalize the ECG signals; the sampling frequency was 500 Hz. A single physician (G.P.) rechecked the ECG recordings in a single-blind manner. The same software was used to calculate the study ECG intervals, as described in detail in previous papers—LabView program (National Instruments, Austin, TX, USA). In particular, the following intervals from the respective time series in ECG recordings were analyzed: R-R mean (RR), Q-R mean (QR), Q-R-S mean (QRS), Q-T mean (QT), S-T mean (ST), and T peak to T end mean (Te) intervals (Figure 1).

To identify the repolarization intervals, we used a software originally proposed by Berger [26] and validated in other subsequent studies [4,5,6,7]. Moreover, we have analyzed the standard deviation (QTe_SD_, QTp_SD_, Te_SD_) values for each of these repolarization phase intervals. Software for data analysis was designed and produced by our research group with the LabView program (National Instruments, Austin, TX, USA).

### 2.3. Statistical Analysis

All variables with normal distribution were expressed as means ± standard deviation, whereas non-normally distributed variables were expressed as median and inter-quartile range (i.r.) and categorical variables as frequencies and percentages (%). Subjects were grouped in 30-day deceased/survivors and positive to Januzzi NT-proBNP cut off/negative. Moreover, mortality, adCHF were analysed in male and female patients separately. One-way ANOVA and Bonferroni tests were used to compare data for the normally distributed variables; Kruskal–Wallis tests were used to compare non-normally distributed variables (as evaluated by the Kolmogorov–Smirnov test); and categorical variables were analyzed with the χ^2^ test. Uni- and multivariable forward (A. Wald) stepwise logistic regression analysis were used to determine the association between mortality or worsening of CHF and other selected clinical and repolarization variables included in the study. In particular, it was considered dependent on variable 30-day mortality and, as covariates, on the following parameters: QTend (QTe) mean, QTpeak (QTp) mean, Tend standard deviation (Te)_SD_ mean, QTend standard deviation (QTe_SD)_, QTpeak standard deviation (QTp_SD)_, Tend standard deviation (Te_SD_)_._ The same method was applied to NT-pro BNP Januzzi cut off with the same ECG variables as covariates. Stepwise multiple regression analysis was used to determine possible relationships between the studied variables. Receiver operating characteristic (ROC) curves were used to determine the sensitivity and specificity of the studied parameters predicting mortality and adCHF as well as areas under curves (AUCs), and 95% confidence intervals (CI) were calculated to compare the diagnostic accuracy in all patients but separately with males and females. All data were evaluated by use of database SPSS-PC+ (SPSS-PC+ Inc., Chicago, IL, USA).

## 3. Results

Considering the initial 239 eligible patients with symptoms of decompensated CHF, 24 patients were excluded because of poor-quality ECGs. Consequently, 215 CHF patients (M/F: 107/108) were finally studied (Table 2).

A total of thirty-four patients died (overall mortality rate, 16%, M/F:17/17), fifteen (7%, M/F: 6/9) died of bronchopneumonia and respiratory failure, fourteen of terminal heart failure (7%, M/F: 9/14), two of fatal myocardial infarction (1%, M/F: 1/1), and three of sudden cardiac death (1%, M/F: 1/2) (of the latter, two of sustained ventricular tachycardia and ventricular fibrillation, and one of acute “cor pulmonale” secondary to a massive embolism). Difference were found regarding causes of death among the two sexes (Table 2).

The general characteristics of the males and females were quite similar (Table 2) except for some data. In particular, women reported significantly higher levels of blood pressures (*p* < 0.05), HDL-cholesterol (*p* < 0.05), and LDL-cholesterol than the male group (Table 2). On the contrary, male patients showed a significant increase in the left ventricular mass index (*p* < 0.001), end-diastolic diameter (*p* < 0.001), and creatinine clearance (*p* < 0.05). Women had a more preserved ejection fraction (*p* < 0.05), while men showed a more frequent history of myocardial ischemia (*p* < 0.001), more premature ventricular complexes (*p* < 0.05), and left (*p* < 0.05) or right (*p* < 0.05) bundle branch blocks. Moreover, men have been implanted with a pacemaker or ICD (*p* < 0.05) more frequently than women (Table 2). Finally, women used statins (*p* < 0.005) and antiplatelet drugs (*p* < 0.05) significantly less often (Table 2).

Regarding the ECG variables, male subjects reported a significant increase in QR (49 ± 21 versus 44 ± 16 ms, *p* < 0.05), QRS (112 ± 35 versus 99 ± 32 ms, *p* < 0.05), QT (479 ± 91 versus 444 ± 75 ms, *p* < 0.05), and ST (366 ± 81 versus 345 ± 61 ms, *p* < 0.001) intervals than female patients. The other examined ECG parameters did not show significant sex differences. Considering all study subjects without sex division, the deceased patients reported a significant increase in Te (121 ± 77 versus 102 ± 26 ms, *p* < 0.001) and Te_SD_ (9i.r6 versus 7i.r.5 ms, *p* < 0.05) in comparison to survival subjects (Figure 2A,C).

Regarding the Te and Te_SD_, males reported a similar trend (Te: 126 ± 23 versus 102 ± 28 ms, *p* < 0.05; TeSD: 9i.r.6 versus 7i.r.5 ms, *p* < 0.05) (Figure 2A,C). On the contrary, deceased women confirmed only the significant increase in Te (117 ± 29 versus 103 ± 26 ms, *p* < 0.05) (Figure 2A), but Te_SD_ (9 i.r. 6 versus 7 i.r 5 ms, *p*: 0.062) did not reach the statistical significance (Figure 2C). Finally, we observed a significant increase in Te and Te_SD_ in men (Te: 111 ± 30 versus 95 ± 23 ms, *p* < 0.05; Te_SD_: 9 interquartile ratio—i.r.—5 versus 5 i.r.2 ms, *p* < 0.001) with higher levels of NT-proBNP (Figure 2B,D). On the contrary, the female patients with higher NT-proBNP levels only showed an increased Te_SD_ (Te: 108 ± 28 versus 100 ± 23 ms, *p*: 0.114; Te_SD_: 9i.r5 versus 5 i.q.2 ms, *p* < 0.001). Male CHF patients with higher NT-proBNP (Jannuzzi cut-off) showed a significant increase in QR_SD_ (6i.r.7 versus 3i.r.5 ms, *p* < 0.05), QRS_SD_ (8i.r.7 versus 5i.r.5 ms, *p* < 0.05), QT (497 ± 98 versus 448 ± 69, *p* < 0.05), QT_SD_ (11i.r.5 versus 7i.r.6 ms, *p* < 0.05), ST (381 ± 91 ms, *p* < 0.05), and ST_SD_ (9i.r. versus 7i.r. ms, *p* < 0.05). On the other hand, the female CHF patients confirmed a significant increase only for QT_SD_ (10i.r.4 versus 7i.r.6, *p* < 0.05) and ST_SD_ (9i.r.4 versus 7i.r.5 ms, *p* < 0.05).

All studied variables obtained from the last part of the repolarization reported a significant specificity-sensitivity curve for mortality (Figure 3A,C,E) and for high levels of NT-proBNP (Figure 3B,D,F).

The NT-proBNP levels were significantly related to Te_SD_ (Figure 4A–C) in both men and women.

## 4. Discussion

The main results of our study demonstrated that male and female patients with CHF have different conditions and parameters to be considered in clinical practice [8,12,13]. As already highlighted in previous studies [8,19], women suffering from heart failure have a substantially more preserved ejection fraction than men (Table 2); this aspect is easily explained considering the CHF physiopathology, characterized by the greater prevalence of hypertensive heart disease, diabetic disorders, and microvessel impairment, rather than the well-known ischemic heart disease affecting the large epicardial coronary vessels among men. For these reasons, some authors have recently felt the need to reevaluate the guidelines and reinterpret the data from the clinical trials in the light of the known clinical and therapeutic differences that HFpEF shows [27]. This demonstrates how the scientific community has understood the importance of evaluating the various clinical phenotypes of heart failure in a more precise and selective manner in order to give increasingly appropriate and equitable responses to the clinical, therapeutic, and quality of life needs of patients, men, and women.

Moreover, from the data collected in our study, women presented higher values of total cholesterol, LDL, and blood pressure. Women frequently reach a late diagnosis [8] with a more severe clinical decompensation [19,28]. In addition, undertreatment largely depends on the erroneous belief that women, even with high cholesterol levels or hypertensive disorders, are protected during their fertile life by estrogens, which, as known, have a protective activity on endothelium and cardiomyocytes. However, the concept of “lipid-load” is increasingly affirming, deserving the clinical community’s attention for a more appropriate use of anti-lipid agents even among young patients [29]. The same underestimation and undertreatment attitude often occur for hypertensive disorders [30]. Conversely, men had a higher left ventricular myocardial index, a sign of more severe hypertensive heart disease, and a better creatinine clearance. Women were significantly older, which was coherent with the epidemiological data in the literature [8].

In order to ensure correct monitoring, even remotely, of patients suffering from heart failure, numerous studies have been produced for the creation of devices and models for risk stratification of exacerbations and acute heart failure [31,32,33,34]. The effort to prevent reacutization must be undertaken using some routinely collected clinical data, such as those derived from ECG repolarization intervals and clinical biomarkers (NT-proBNP).

The Tend mean and the T end standard deviation have been shown to be effective in predicting the mortality of patients hospitalized for acute heart failure in various associated pathological conditions [3,4,5,7,35]. Moreover, they seem to be useful for intercepting those patients who deserve higher attention for progressive hemodynamic lability [3,4,5,7,34]. In our study, an accurate evaluation was carried out for disaggregated data by sex, which was guaranteed to highlight how the Tend SD, and not the Tend mean, is a reliable parameter in the female population for predicting mortality and reacutization in CHF patients.

These data will therefore be carefully considered in the creation of risk models as specific sex markers for specific cardiovascular preventions [36]. The use of electrocardiographic variables that take into consideration the ventricular repolarization marker for early risk stratification of adCHF patients is a promising field of clinical research [3,4,5]. In fact, together with the evaluation of NT-proBNP and the use of eHealth, artificial intelligence, and machine learning tools, it could be possible to produce predictive models with these inexpensive, easily available parameters for patient monitoring.

## 5. Study Limitations

An actual limitation of the study is the absence of patients treated with SGLT2 inhibitors. The sample was, in fact, largely studied before the recent indications provided by the European Society of Cardiology guidelines [25] on the use in class I evidence A of these drugs in subjects with HF and diabetes, hence the impact on repolarization, frequency of reacutizations, and clinical outcomes. Further enrollment in this registry will help fill this gap. Moreover, an interventional study could definitively assess the power and utility of such evidence.

## Figures and Tables

**Figure 1 jcm-12-04714-f001:**
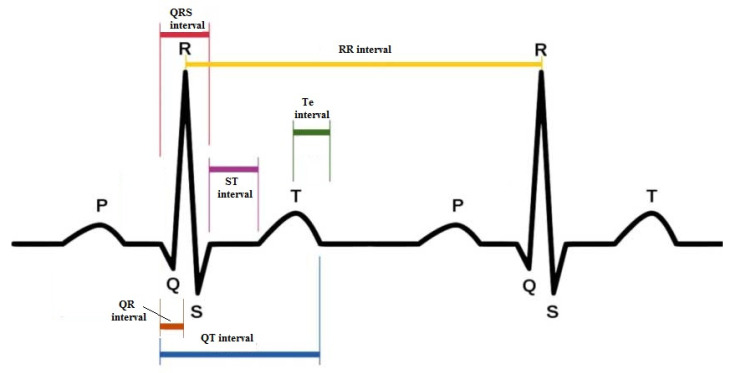
Intervals observed and analyzed during the 5 min ECG recordings.

**Figure 2 jcm-12-04714-f002:**
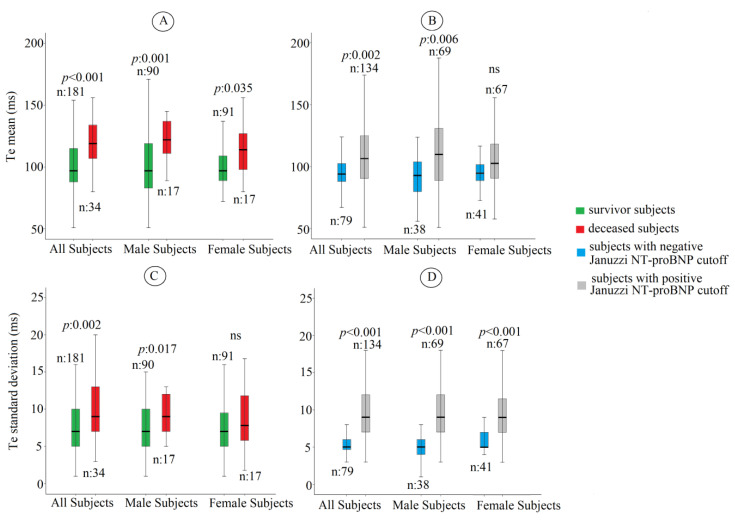
Te mean and Tend SD for survived patients and deceased ones (**A**,**C**) and for subjects with increased and normal levels of NT-proBNP (**B**,**D**).

**Figure 3 jcm-12-04714-f003:**
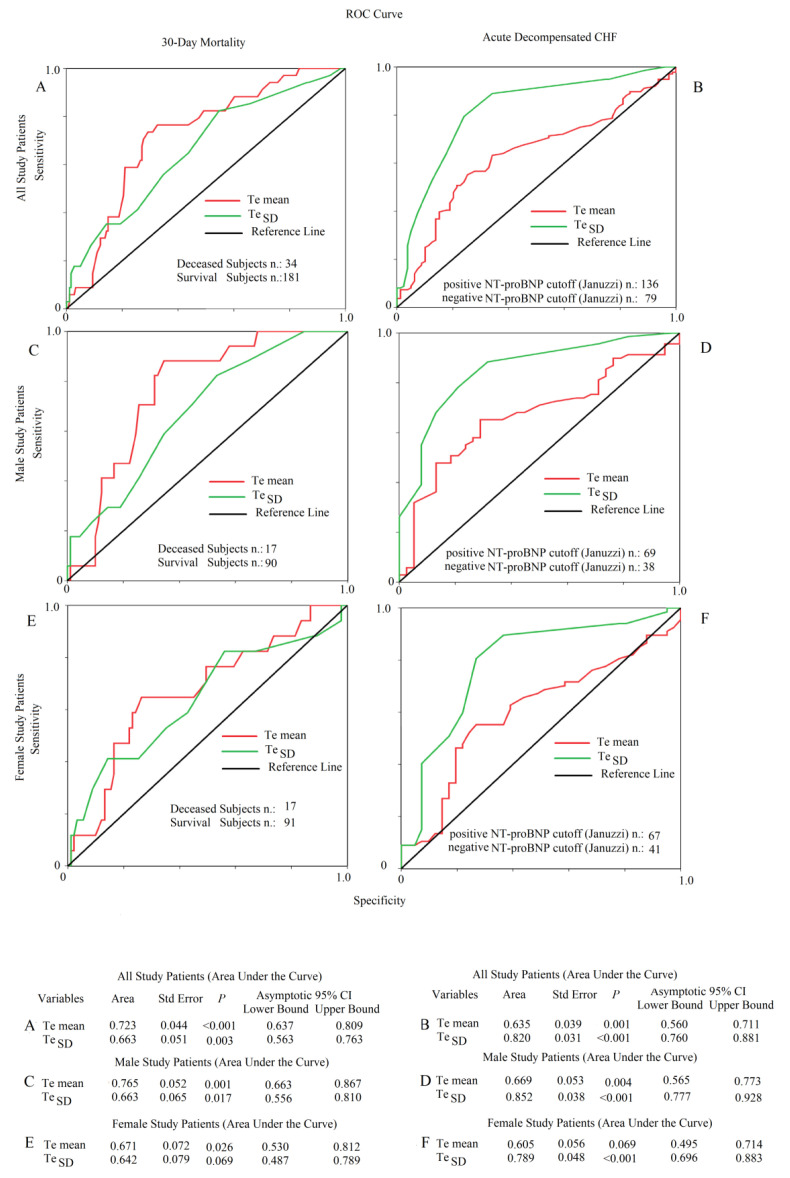
Receiving operating curves for Te mean and Te standard deviation for mortality based on NT-proBNP levels. (**A**) Mortality among all patients for Te mean and Te Standard Deviation (SD) markers. (**B**) Heart failure decompensation related to NT-proBNP levels among all patients for Te mean and Te SD markers. (**C**) Mortality among male patients related to to Te mean and TeSD. (**D**) Heart failure decompensation among male patients related to Te mean and TeSD markers. (**E**) Mortality among female patient related to Te mean and Te SD markers. (**F**) Heart failure decompensation among female patients related to Te mean and TeSD markers.

**Figure 4 jcm-12-04714-f004:**
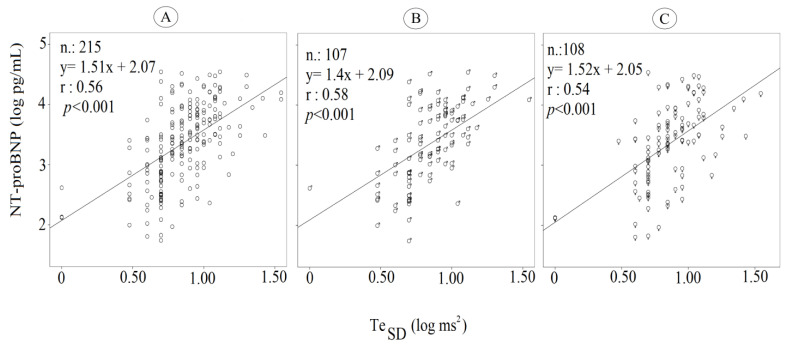
NT-proBNP and Te_SD_ correlations in all patients (**A**), among men (**B**), and among women (**C**).

**Table 1 jcm-12-04714-t001:** Peculiarities of chronic heart failure among women.

Specific Characteristics of Chronic Heart Failure in Women	
Pathophysiology	- Endotelial dysfunction, microvessel damage (diabetes, arterial hypertension, estrogens depletion after menopause, etc.)
Symptoms	- More severe weakness, reduced exercise tolerance, diaphoresis, more pronounced dyspnea, precordial palpitations
Diagnostic delay determing a later-in-life diagnosis	- High degree of polypathology, polypharmacy, and iatrogenic damage - Reduced access to the heart transplant - Exclusion from clinical trials
Difficult prognostic evaluation	- Scores/risk charts formulated on male models, and therefore not designed and studied for women - No score currently takes into account sex-specific risk factors.
High “revolving door” risk	- High costs for the health systems - Reduced patients’ quality of life

**Table 2 jcm-12-04714-t002:** General Characteristics of the Study Subjects.

	All Subjects	Male Subjects	Female Subjects	*p*
	N:215	N:107	N:108	
Age, years	84 ± 8	81 ± 10	85 ± 9	**0.003**
BMI, kg/m^2^	24.3 ± 3.0	26.3 ± 5.0	25.3 ± 4.7	0.131
Heart Rate, beats/m	76 ± 15	73 ± 14	76 ± 12	0.214
Systolic Blood Pressure, mm Hg	127 ± 18	122 ± 19	129 ± 19	**0.010**
Diastolic Blood Pressure, mm Hg	70 ± 11	68 ± 10	72 ± 11	**0.005**
Left Ventricular Ejection Fraction, %	44 ± 9	40 ± 11	44 ± 10	**0.009**
Left Ventricular Mass Index, g/m^2^	143 ± 29	145 ± 36	127 ± 31	**<0.001**
Left Ventricular End-Diastolic Diameter, mm	53 ± 6	56 ± 7	50 ± 7	**<0.001**
Posterior Wall Thickness, mm	12 ± 1	11 ± 2	11 ± 1	0.994
Interventricular Septum Thickness, mm	12 ± 2	12 ± 1	12 ± 1	0.629
Left Atrial Transversal Diameter, mm	47 ± 7	48 ± 6	46 ± 7	0.103
Tricuspid Annular Plane Systolic Excursion, mm	21 ± 5	20 ± 4	20 ± 4	0.956
Tricuspid Regurgitation Peak Gradient, mmHg	44 ± 15	44 ± 15	44 ± 12	0.836
Heart Failure with Reduced Ejection Fraction, n(%)	98(46)	58(54)	40(37)	**0.011**
Heart Failure with Mildy Reduced Ejection Fraction, n(%)	32(15)	13(12)	19(18)	0.262
Heart Failure with Preserved Ejection Fraction, n(%)	85(40)	36(34)	49(45)	0.079
NT-pro BNP, pg/mL	2895[7350]	2951[8235]	2865[6489]	0.886
C-reactive protein (mg/dl)	3.7[8.7]	3.4[8.4]	4.4[9.0]	0.127
High sensitivity cardiac troponin/(pg/L)	42[59]	43[60]	41[59]	0.365
Creatinine clearance (mL/m)	44 ± 25	53 ± 29	44 ± 31	**0.039**
Fasting glucose (mmol/L)	6.46 ± 1.93	6.48 ± 2.25	7.12 ± 2.46	0.057
HbA_1c_ (%)	6.01 ± 1.20	5.87 ± 1.20	6.40 ± 1.49	**0.010**
Total Cholesterol (mmol/L)	3.70 ± 1.00	3.55 ± 1.01	3.85 ± 1.05	0.119
HDL-cholesterol (mmol/L)	1.13 ± 0.42	1.02 ± 0.33	1.15 ± 0.40	**0.048**
LDL –cholesterol (mmol/L)	1.99 ± 0.81	1.85 ± 0.81	2.16 ± 0.84	**0.041**
Triglycerides (mmol/L)	1.67 ± 1.49	1.80 ± 1.34	1.56 ± 0.91	0.254
Hypertension, n (%)	166(77)	80(75)	86(80)	0.395
Hypercholesterolemia, n (%)	98(46)	55(51)	43(40)	0.088
Diabetes, n (%)	90(42)	44(41)	46(43)	0.827
Renal Insufficiency, n (%)	105(49)	57(53)	48(44)	0.195
Known Myocardial Ischemia History, n (%)	75(35)	51(48)	24(22)	**<0.001**
Premature Supraventricular Complexes, n (%)	20(9)	14(13)	6(6)	0.057
Premature Ventricular Complexes, n (%)	48(22)	30(28)	18(17)	**0.045**
Permanent Atrial fibrillation, n (%)	76(35)	37(35)	39(36)	0.814
Left Bundle Branch Block, n (%)	44(21)	28(26)	16(15)	**0.039**
Right Bundle Branch Block, n (%)	33(15)	24(22)	9(8)	**0.004**
Pacemaker-ICD, n (%)	48(22)	32(30)	16(15)	**0.008**
Deceased Subjects, n (%)	34(16)	17(16)	17(16)	0.976
β-blockers, n (%)	144(67)	73(68)	71(66)	0.699
Furosemide, n (%)	166(77)	84(79)	82(76)	0.652
ACE/Sartans	87(41)	44(41)	43(40)	0.845
Aldosterone antagonists, n (%)	31(14)	14(13)	17(16)	0.579
Potassium, n (%)	15(7)	5(5)	10(9)	0.187
Nitrates, n (%)	28(13)	14(13)	14(13)	0.979
Digoxin, n (%)	10(4)	6(6)	4(4)	0.507
Statins, n (%)	64(30)	42(39)	22(20)	**0.002**
Antiplatelet drugs, n (%)	83(39)	51(48)	32(30)	**0.007**
Oral Anticoagulants, n (%)	61(28)	27(25)	34(32)	0.310
Diltiazem or Verapamil, n (%)	7(3)	2(2)	5(5)	0.254
Dihydropyridine Calcium channel blockers, n (%)	28(13)	13(12)	15(14)	0.705
Propafenone, n (%)	2(0.9)	0(0)	2(1.9)	0.157
Amiodarone, n (%)	19(9)	11(10)	8(7)	0.458
Valsartan/Sacubitril, n (%)	4(1.9)	2(2)	2(2)	0.993

Data are expressed as mean ± SD, or median [interquartile range], or number of patients (%)

## Data Availability

All data, materials, and codes used in this study are available upon request from the corresponding author.

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
