# Peer review of "Sex Differences in Repolarization Markers: Telemonitoring for Chronic Heart Failure Patients"

_jcm, 2023, doi:10.3390/jcm12144714_

Round 1

Reviewer 1 Report

The manuscript titled “Sex differences in repolarization markers: telemonitoring for chronic heart failure patients” compared the ECG makers between male and female CHF patients and identified TeSD could be a promising maker for CHF worsening and complications. To investigate the sex difference in ECG makers is novel and interesting; however, provided evidence is not enough to support their hypothesis, and some of the interpretation of results is not correct. Moreover, the objective and the conclusion are not consistent since the aim is to find the different ECG markers in male and female CHF patients, but the conclusion states that TeSD is a risk factor for all patients.

Specific comments:

1. For the abstract, the results for repolarization markers between males and females, like the sentence “Data for Te mean and TeSD for survived patients and deceased ones and for subjects with increased and normal levels of NT-proBNP showed that both of them were increased among deceased and decompensated men,” are confusing,

2. For Table 2, the p-value for Heart Failure with Preserved Ejection Fraction is 0.079, but Line 149 shows ‘Women had a more preserved ejection fraction (p<0.05)’. The inconsistency in the data raises questions about the accuracy of the results.

3. The legend for Figure 4 is missing.

Moderate editing of English language required

Author Response

#Reviewer 1

We would like to thank the reviewer for his/her time spent for the revision. We have done our best to fulfill the requests.

The manuscript titled “Sex differences in repolarization markers: telemonitoring for chronic heart failure patients” compared the ECG makers between male and female CHF patients and identified TeSD could be a promising maker for CHF worsening and complications. To investigate the sex difference in ECG makers is novel and interesting; however, provided evidence is not enough to support their hypothesis, and some of the interpretation of results is not correct. Moreover, the objective and the conclusion are not consistent since the aim is to find the different ECG markers in male and female CHF patients, but the conclusion states that TeSD is a risk factor for all patients.

Reply: We thank the reviewer for the comment which gives us the opportunity to better explain the findings of our study. We initially aimed to evaluate differences in known markers of ventricular repolarization, used for risk stratification for mortality and worsening of chronic heart failure, between men and women. Te mean was highr in both sexes among deceased patients, while TeSD was increased among decompensated subjects (higher NT proBNP levels)in both sexes.

Reasoning on the clinical applicability of these data, and since TeSD proved to be a reliable marker of heart failure exacerbation in both men and women, we proposed the use of this marker and not others for clinical and research setting to find earlier high risk patients.

Specific comments:

  1. For the abstract, the results for repolarization markers between males and females, like the sentence “Data for Te mean and TeSD for survived patients and deceased ones and for subjects with increased and normal levels of NT-proBNP showed that both of them were increased among deceased and decompensated men,” are confusing,

Reply: you are right, so we changed the sentence as follow “. Data for Te mean showed increased values among deceased men and women compare to survivals, but TeSD showed to be the most reliable marker for CHF reacutization in both sexes.”

  1. For Table 2, the p-value for Heart Failure with Preserved Ejection Fraction is 0.079, but Line 149 shows ‘Women had a more preserved ejection fraction (p<0.05)’. The inconsistency in the data raises questions about the accuracy of the results.

Reply: Thanks for your remark. We assert that the ejection fraction of women is more conserved than that of men because again in table two under the heading Left Ventricular Ejection Fraction the reported statistical significance is p value 0.009. There is actually no numerical difference between patients who have preserved ejection fraction (number of patients with EF>50%), but among those who have reduced ejection traction, women still have a fraction of ejection tends to be more preserved. This supports the results of our study.

  1. The legend for Figure 4 is missing.

Reply: the caption has been added.

Reviewer 2 Report

Overall, the authors provide a clear overview of the study's background, aim, methodology, and key results. It highlights the importance of considering sex differences in ECG repolarization markers for CHF patients undergoing telemonitoring and suggests Te SD as a potential risk factor.

Line 56: As evidenced by many recent studies [8, 9, 10, 11, 12, 13, 14], heart failure in 56 women shows extremely peculiar characteristics.

The listed prior works need to be addressed specifically.

Line 57: The physiopathology of the damage is 57 strictly related to the endothelial dysfunction and microvessels damage [15, 16, 17] 58 which lead to a CHF with a more preserved ejection fraction than in men [8]. For this 59 reason, symptoms, clinical course, therapeutic response and prognosis [18, 19, 20, 21] 60 (table 1) are peculiar as well as the repolarization ECG patterns [22, 23] .

The same improvement is needed as above

Line 102: If Figure 1 is adapted from the third party, please give out the renference. The caption of Figure 1 is "Intervals obtained from 5 min ECG recording." If this graphic is not exacted from the collected data, the caption needs change too.

Line 25: Stepwise multiple regression analysis 125 was used to determine possible relationships between the studied variables.

What is the final model selected by stepwise variable selection?

The quality of English language in the provided manuscript is fair, but there are some areas that could be improved for clarity and readability. Here are a few suggestions:

  1. Use clear and concise language: Some sentences in the text are long and complex, which can make them difficult to follow. Try breaking them down into shorter sentences to improve readability.

  2. Maintain consistent verb tenses: Ensure that the verb tenses are consistent throughout the text. For example, in the sentence "This data will therefore be carefully considered," the verb tense should be consistent with the rest of the text, which predominantly uses the past tense.

  3. Watch out for word repetition: There are instances where the same words or phrases are repeated within close proximity. Try using synonyms or rephrasing to avoid unnecessary repetition.

  4. Check sentence structure: Review the sentence structure to ensure clarity and coherence. In some cases, the sentence structure can be improved by rearranging or rephrasing certain phrases.

  5. Proofread for spelling and punctuation: Double-check for any spelling or punctuation errors that may have been overlooked. This will help improve the overall polish and professionalism of the writing.

By implementing these suggestions, authors can enhance the clarity and flow of the text, making it easier for readers to understand.

Author Response

#Reviewer 2

We would like to thank the reviewer for his/her time spent for the revision. We have done our best to fulfill the requests.

  • Overall, the authors provide a clear overview of the study's background, aim, methodology, and key results. It highlights the importance of considering sex differences in ECG repolarization markers for CHF patients undergoing telemonitoring and suggests Te SD as a potential risk factor.

Reply: thank you for your appreciation.

  • Line 56: As evidenced by many recent studies [8, 9, 10, 11, 12, 13, 14], heart failure in 56 women shows extremely peculiar characteristics.

              The listed prior works need to be addressed specifically.

Reply: we have added a specific paragraph, addressing the studies as requested.

  • Line 57: The physiopathology of the damage is 57 strictly related to the endothelial dysfunction and microvessels damage [15, 16, 17] 58 which lead to a CHF with a more preserved ejection fraction than in men [8]. For this 59 reason, symptoms, clinical course, therapeutic response and prognosis [18, 19, 20, 21] 60 (table 1) are peculiar as well as the repolarization ECG patterns [22, 23] .

               The same improvement is needed as above

Reply: you are right and we understand your suggestion, but we have added just a sentence in this case, but we prefer to not get the introduction longer, if you agree. We have already put some specific explanations in the table 1 in order to resume the state of the art in this topic.

  • Line 102: If Figure 1 is adapted from the third party, please give out the renference. The caption of Figure 1 is "Intervals obtained from 5 min ECG recording." If this graphic is not exacted from the collected data, the caption needs change too.

Reply: the figure is originally elaborated from our graphic expert. The caption has been modified with ”observed and analyzed” in place of “obtained”. Thank you.

  • Line 25: Stepwise multiple regression analysis 125 was used to determine possible relationships between the studied variables.

               What is the final model selected by stepwise variable selection?

Reply: linear regression model was used (as reported in figure 4) between proBNP and Tend SD data.

Comments on the Quality of English Language

The quality of English language in the provided manuscript is fair, but there are some areas that could be improved for clarity and readability. Here are a few suggestions:

  1. Use clear and concise language: Some sentences in the text are long and complex, which can make them difficult to follow. Try breaking them down into shorter sentences to improve readability.
  2. Maintain consistent verb tenses: Ensure that the verb tenses are consistent throughout the text. For example, in the sentence "This data will therefore be carefully considered," the verb tense should be consistent with the rest of the text, which predominantly uses the past tense.
  3. Watch out for word repetition: There are instances where the same words or phrases are repeated within close proximity. Try using synonyms or rephrasing to avoid unnecessary repetition.
  4. Check sentence structure: Review the sentence structure to ensure clarity and coherence. In some cases, the sentence structure can be improved by rearranging or rephrasing certain phrases.
  5. Proofread for spelling and punctuation: Double-check for any spelling or punctuation errors that may have been overlooked. This will help improve the overall polish and professionalism of the writing.

By implementing these suggestions, authors can enhance the clarity and flow of the text, making it easier for readers to understand.

Reply: thank you very much for your precious suggestions, we tried to follow them ameliorationg the text and correcting the typos..

Round 2

Reviewer 1 Report

Thank you for the author's responses. I do not have any further comments.